# Thymic Epithelial Tumor and Immune System: The Role of Immunotherapy

**DOI:** 10.3390/cancers15235574

**Published:** 2023-11-25

**Authors:** Matteo Perrino, Nadia Cordua, Fabio De Vincenzo, Federica Borea, Marta Aliprandi, Luigi Giovanni Cecchi, Roberta Fazio, Marco Airoldi, Armando Santoro, Paolo Andrea Zucali

**Affiliations:** 1Department of Oncology, IRCCS Humanitas Clinical and Research Center, Via Manzoni 56, Rozzano, 20089 Milan, Italy; matteo.perrino@cancercenter.humanitas.it (M.P.); nadia.cordua@cancercenter.humanitas.it (N.C.); fabio.de_vincenzo@cancercenter.humanitas.it (F.D.V.); armando.santoro@cancercenter.humanitas.it (A.S.); 2Department of Biomedical Sciences, Humanitas University, Via Rita Levi Montalcini 4, Pieve Emanuele, 20090 Milan, Italy; federica.borea@cancercenter.humanitas.it (F.B.); marta.aliprandi@cancercenter.humanitas.it (M.A.); luigi.cecchi@cancercenter.humanitas.it (L.G.C.); roberta.fazio@cancercenter.humanitas.it (R.F.); marco.airoldi@cancercenter.humanitas.it (M.A.)

**Keywords:** thymic epithelial tumors, thymoma, thymic carcinoma, immune tolerance, immune checkpoint inhibitors, cancer vaccines, immune-related adverse events, biomarkers

## Abstract

**Simple Summary:**

Similarly to other solid malignancies, there is a growing interest in testing immune checkpoint inhibitors (ICIs) for thymic epithelial tumors (TETs). However, the thymus has unique biological features that can increase the risk of immune-related adverse events. In this article, the relationships among the thymus, immune system, and autoimmune diseases (ADs) usually associated with TETs, and their impact on the potential benefits and risks of immunotherapy, are reviewed. The results of completed clinical trials of immunotherapy for refractory/relapsed thymomas and thymic carcinomas are presented.

**Abstract:**

Thymic epithelial tumors (TETs) comprise a rare group of thoracic cancers, classified as thymomas and thymic carcinomas (TC). To date, chemotherapy is still the standard treatment for advanced disease. Unfortunately, few therapeutic options are available for relapsed/refractory tumors. Unlike other solid cancers, the development of targeted biologic and/or immunologic therapies in TETs remains in its nascent stages. Moreover, since the thymus plays a key role in the development of immune tolerance, thymic tumors have a unique biology, which can confer susceptibility to autoimmune diseases and ultimately influence the risk–benefit balance of immunotherapy, especially for patients with thymoma. Indeed, early results from single-arm studies have shown interesting clinical activity, albeit at a cost of a higher incidence of immune-related side effects. The lack of knowledge of the immune mechanisms associated with TETs and the absence of biomarkers predictive of response or toxicity to immunotherapy risk limiting the evolution of immunotherapeutic strategies for managing these rare tumors. The aim of this review is to summarize the existing literature about the thymus’s immune biology and its association with autoimmune paraneoplastic diseases, as well as the results of the available studies with immune checkpoint inhibitors and cancer vaccines.

## 1. Introduction

Thymic epithelial tumors (TETs) are the most common neoplasms of the anterior mediastinum, with a worldwide incidence of 1.3–3.2 cases per million/year [1]. The mean age at diagnosis is 50–60 years. The World Health Organization (WHO) histopathological classification categorizes TETs as thymoma, further divided into types A, AB, B1, B2, and B3, and thymic carcinoma (TC) [2]. Compared to thymoma (incidence rate 2.8 per million), TCs are extremely rare (incidence < 0.1 per million) but more aggressive due to their lymphatic and hematogenous spread.

Independent prognostic factors for TETs include histology, stage, and completeness of surgical resection. Types A, AB, and B1 present an overall survival (OS) rate of more than 82–100% at 10 years, whereas for types B2 and B3 and thymic carcinomas, the OS is 87%, 64%, and 27%, respectively [3,4,5]. The completeness of resection is very important for prognosis, even for stages III and IV tumors. Indeed, the 5-year survival rates of completely resected patients resulted in an OS of 92%, 94%, 83%, and 76% for stages I, II, III, and IV, respectively [5,6,7,8].

The management of TETs primarily relies only on retrospective analyses, prospective single-arm trials, and experts’ opinions. Therefore, according to all the international guidelines, TETs require a multidisciplinary approach and a specific expertise to improve outcomes and reduce long-term treatment-related sequelae [9,10,11,12]. Today, surgery is the main treatment for patients with resectable disease. However, 10–30% of patients relapse even after 10 to 20 years from a radical resection [13]. With the aim being to increase the resectability rate and to reduce the risk of disease relapse, perioperative treatments (chemotherapy and/or radiotherapy) should be considered for patients with local advanced or unresectable disease [14,15]. In unresectable locally advanced or metastatic disease, the current standard of care consists of platinum-based chemotherapy in combination with anthracycline (CAP or ADOC regimens), or with etoposide or paclitaxel for patients who are unfit for anthracyclines. Unfortunately, in the palliative setting, chemotherapy exhibits limited effectiveness, and there is a lack of conclusive evidence regarding treatment options beyond first-line therapy [11,16,17]. Moreover, the process of identifying new therapeutic targets is slow due to the rarity of TETs and the difficulty in enrolling patients in phase II–III trials. Promising results have exclusively been achieved in small phase II studies with cKIT inhibitors (Imatinib), mTOR inhibitors (Everolimus), and anti-angiogenic agents (Sunitinib and Lenvatinib) [18,19,20,21,22].

Considering the great levels of Programmed Death Ligand 1 (PD-L1) expression in TETs, ranging from 23% to 92% for thymomas and from 34% to 88% for TCs, Programmed Death 1 (PD-1)/PD-L1 inhibitors have shown a very promising clinical activity [23,24]. PD-L1 is an immune-inhibitory molecule that, in binding its receptor (PD1) on immune cells, suppresses T cell activation and promotes carcinogenic activity and tumor progression. However, TETs are often associated with autoimmune diseases (ADs), justifying a higher degree of immune-related adverse events (irAEs) observed during immunotherapy [25]. In fact, 40% of patients with thymoma and 6% in patients with TC present Myastenia Gravis (MG) or other ADs. Our knowledge of the biological basis regulating the relationships among immune system, TETs, and systemic autoimmune diseases still remains unclear.

This review aims to present the knowledge about the biological relationship among the thymus, immune system, and ADs associated with TETs, and report the currently available evidence about immunotherapy in TETs, in order to analyze the balance between expected benefits and potential toxicities.

## 2. Materials and Methods

We conducted a non-systematic literature review on autoimmune disorders and the state of the art of immunotherapy in TETs. We explored PubMed and Scopus databases to select publications from 2009 up to July 2023. A combination of the following keywords was used for a title abstract search: “Thymus”, “Thymic Epithelial Tumor”, “Thymoma”, “Thymic Carcinoma”, “Immune System”, “Autoimmunity”, “Autoimmune Disorders”, “Paraneoplastic Disorders”, “Immunotherapy”, “Programmed cell death-1”, “Programmed cell death-ligand 1”, “Myasthenia Gravis”, “Anticancer Treatment”, “Thoracic Surgery”. Only English articles were included.

## 3. Immune System and Thymus

The central role of the thymus in regulating adaptive immunity is supported by an organized structure and specialized cellular microenvironment (Figure 1).

The thymus comprises a cortical and medullary region, each responsible for distinct phases of T cell maturation. In addition to T cell precursors, thymic microenvironment is composed of different cell types including endothelial cells, mesenchymal cells, macrophages, dendritic cells, and thymic epithelial cells (TECs). TECs are specialized in two further subsets, cortical and medullary TECs (cTECs and mTECs, respectively), which localize to their corresponding region [26,27,28].

The selection of T cells in the thymus begins when circulating lymphoid progenitor cells migrate into thymic cortico-medullary junction. Here, the interaction of the lymphoid precursor with the Notch ligand commits cells to the T lineage [29]. At this phase of development, early T cells lack the expression of both CD4 and CD8 and are referred as double-negative (DN) thymocytes. Once they have entered the gland, T cell precursors reach the cortical region via the expression of chemokine receptors (CCR9 and CCR7) following the gradients of their corresponding ligands (CCL21 and CCL25) produced by stromal cells [30]. In the cortical region, DN thymocytes initiate the rearrangement of the T cell receptor (TCR) locus of the β chain. Only functional β chain-expressing cells survive this selection, thus preventing cells with out-of-frame rearrangement from further development. Once selected, T cells simultaneously express CD4 and CD8, becoming double-positive (DP) thymocytes. At this phase of T cell development, the α-chain locus of the TCR begins recombination, eventually resulting in a mature αβ-TCR [31].

At this point, DP thymocytes undergo positive selection. Functional TCR expressed on DP thymocytes interacts with major histocompatibility complexes (MHC) expressed by cTEC. Clones of thymocytes that are able to recognize antigen on the MHC receive a survival stimulus. On the other hand, thymocytes that fail to recognize self-MHC undergo apoptosis, also known as death by neglect [32,33].

In addition to positive selection, the type of MHC recognized by developing thymocytes is responsible for lineage commitment. According to the class of MHC recognized during positive selection (MHC-I or MHC-II), double-positive thymocytes commit to either CD8 or CD4 lineage, becoming single positive (SP) thymocytes. Recognition of the MHC-II activates ZBTB7B transcription factor establishing a gene expression program that differentiates activated T cells into CD4+ helper cell at the periphery. On the other hand, recognition by thymocytes of the MHC-I activates RUNX3, resulting in a cytotoxic CD8 T cell phenotype upon cellular activation [29,34,35].

Following completion of positive selection, SP thymocytes overexpress CCR7 and CCR4 and migrate in the thymic medulla following CCL19 and CCL21 chemokines secreted by mTECs [36,37].

In the medulla, SP thymocytes undergo negative selection, in which self-reactive T cells are removed. In order to ensure the expression of tissue specific self-antigens (TSAs) beyond anatomical and developmental restrictions of the thymus, the medulla is equipped with specialized antigen presenting cells (APC), mTECs. mTEC are characterized by autoimmune regulator (AIRE), a transcription factor that controls the expression of genes not limited to thymic cells. Therefore, during negative selection, T cells are tested for self-reactivity of TCR beyond thymus-specific antigens [38,39]. Besides AIRE, other genes have been shown to contribute to mTEC expression of ectopic TSAs, such as FEZF2 [40].

During negative selection, the high-affinity interaction between TCR and self-peptide-MHC, in the presence of costimulatory signals (CD28), activates the apoptotic cascade through BIM, culminating in cell death [35,41].

However, not all self-reactive T cells undergo to apoptosis during negative selection. Indeed, a fraction of CD4 thymocytes with strong MHC-II interactions transform into regulatory T cells (Treg). Tregs are characterized by selective expression of transcription factor FOXP3 and play a central role in establishing immune tolerance. Such immunomodulatory function has been confirmed by the observation of autoimmune and immune-mediated disorders associated with Treg depletion or dysfunction in animal models and humans [42].

Multiple factors have been shown to interplay in defining the lineage of self-reactive CD4 thymocytes into Treg. Specifically, IL-2 rescues self-reactive cells upon TCR interaction by increasing the expression of CD25. The interaction of IL-2 with its receptor (CD25) promotes the expression of FOXP3 resulting in the Treg phenotype [35,42].

The collective steps of negative selection and Treg cells development establish central tolerance, among the primary strategies in preventing autoimmunity.

## 4. Autoimmune Diseases and TETs

One third of patients with TETs present a paraneoplastic ADs [43]. The incidence of ADs is 6% in TC, but almost 40% in thymoma [44], and varies in relation to the histologic subgroup, >40% in B2, 40% in B3, and 26% in B1 [25].

The most common AD is MG, a neuromuscular junction disease characterized by fatigable muscle weakness. The symptoms are variable according to affected musculoskeletal districts. MG should be suspected in patients presenting with dysphagia and dysarthria (bulbar symptoms), asymmetric eyelid ptosis and diplopia, respiratory weakness, and persistent cough [45]. A proportion of 24.5–44% of patients with TETs develop MG at the time of diagnosis or subsequently; conversely, 15–20% of patients with MG have a thymoma. Almost 90% of MG patients have anti-acetylcholine receptor antibodies (AChR-Abs), whereas the others present, in 25–47% of cases, anti-muscle-specific tyrosine kinase antibodies (Musk-Abs) [46,47]. The diagnosis of MG is confirmed through antibody testing and electromyography studies showing a post-synaptic neuromuscular transmission deficit.

Other paraneoplastic ADs associated with TET are Pure Red Cell Aplasia (PRCA), hypo-gammaglobulinemia (Good syndrome), systemic lupus erythematous and polymyositis with a prevalence of 1–5% of cases, as well as immunodeficiency [9,48,49,50].

The PRCA is a rare anemia secondary to the failure of erythropoiesis caused by an absence of red blood cell precursors in the bone marrow. The incidence is 2–5% in patients with thymoma, whereas TET can be detected in 10–20% of patients with PRCA. It is characterized by normochromic, normocytic anemia associated with absent or low number erythroblasts in the bone marrow and reticulocytopenia in the peripheral blood. The immune-mediated erythropoietic failure is crucial to PRCA in patients with TETs and laboratory evidences have shown the presence of both cell-mediated and antibody-mediated mechanisms of inhibition of red cell erythropoiesis [51].

The Good syndrome is an uncommon disorder which involves both deficient humoral and cellular immunity causing an increased risk of opportunistic infections and second malignancies. The incidence is in the range of 3–6% of patients with thymoma, and its pathogenesis is unknown. Immunodeficiency is associated with the presence of anti-cytokine autoantibodies [50,52].

Patients with one AD could present additional immune-mediated conditions [53]. In a systematic review of case reports evaluating 507 patients with thymoma and 123 different ADs, Zhao et al. detected MG in 63% of cases, PRCA in 7.7%, lichen planus in 6.3%, Good syndrome in 5.9%, and limbic encephalitis in 5.9%. Moreover, they identified two or more ADs in 49% of all thymoma patients [53]. In another analysis including 302 patients with TETs, Benitez et al. observed ADs in 12.5% of cases, and 4.8% of patients had two or more ADs [54].

In general, paraneoplastic ADs in TETs can develop at any time, before or after tumor diagnosis, and even after thymectomy [43,49,52,54].

The pathophysiological links between TETs and autoimmunity as well as the dichotomy between thymoma and TC in their co-morbidity with ADs are completely unknown [49].

Several theories have been postulated to explain the pathogenic mechanisms associating TETs and AD and they all speculate on possible failures of positive and negative thymic selections of T-lymphocytes in adulthood. In particular, the “escape theory” hypothesizes that immature thymoma-derived T cells escape the altered tumor environment without undergoing the negative selection in the medulla where self-tolerance is induced [43]. The “genetic theory” suggests that the uncontrolled proliferation of auto-reactive thymoma clones can overcome the capacity of thymus selection. The “AIRE theory” implies decrease expression of AIRE (lost in 95% of thymomas), FEZF2, and MHC class II in malignant TECs causing a dysfunctional antigen presentation. AIRE deficiency can lead to an impaired positive selection of immunosuppressive regulatory T cells (Tregs) and auto-reactive T cells can be released from the thymus, thus predisposing to autoimmunity [43,55,56]. However, the finding that ADs can develop after thymectomy suggest a more complex defect in T-lymphocyte maturation [56,57]. In a retrospective study, Bernard et al. observed the development of an AD after thymectomy in up to 8% of patients. Moreover, 2% of patients had an AD before or concomitantly to thymectomy and develop another AD, while 15% of patients had not previous had an AD. The authors did not identify any risk factor associated with AD development after thymectomy, and in particular, they did not find a correlation between a prior AD and the risk of an AD after surgery. This study also demonstrated that thymectomy did not have an effect on the natural history of ADs in TET patients: even if remission of MG was obtained in 93% of patients after thymectomy, a relapse was observed in 41% of cases [48,49]. A possible explanation for this could be that patients with an AD requiring thymectomy may have a predisposition to autoimmunity and, thus, to developing other ADs in the future. Another hypothesis could be that a thymic abnormality, especially a neoplastic disease, definitely alters the construction of the immune repertoire, so predisposing, even after thymectomy, to ADs. In a systemic review of thymoma patients who underwent radical thymectomy, Zhao et al. observed ADs remission in 76% of cases, which subsequently recurred or developed a new AD in 21% of cases (in a time from 1 month to 19 years after surgery) [53]. Moreover, 18% of patients had persisting AD after thymectomy, and among these patients, 40% developed an additional AD. In this analysis, the complete resolution of the associated paraneoplastic AD was an independent predictive factor of a better OS.

The presence of ADs seems to be associated with favorable prognostic risk factors. Padda et al. observed that patients with TET and ADs had a significantly better relapse-free survival (RFS) (10-year RFS in ADs+ TETS 17.3% vs. 21.2% in ADs- TETs, *p* = 0.0003) and OS (median OS in AD+ TETS 21 years vs. 17 years in ADs-, *p* < 0.0001) than those without ADs [44]. Similarly, Filosso et al. reported improved OS in patients with TET and MG compared with those without MG [58]. A potential explanation could be that TETs + ADs are usually diagnosed at an earlier stage due to the presence of symptoms leading to higher rates of complete resection [5].

## 5. Role of Immunotherapy

### 5.1. Immune Checkpoint Inhibitors

Both thymomas and TCs commonly express high levels of PD-L1 [59,60,61]. Immune checkpoint inhibitors (ICIs) have shown an interesting clinical activity, although current evidence is limited to early phase trials with small sample size of pretreated patients.

Pembrolizumab and Nivolumab were tested in three phase II trials, having the overall response rate (ORR) as the primary endpoint and enrolling patients previously treated with at least one line of platinum-based chemotherapy (Table 1).

In a phase II trial, Pembrolizumab was administered in 33 patients (7 thymomas; 26 TCs) achieving partial response (PR) in 2/7 patients with thymomas (28.6%) and in 5/26 patients with TCs (19.2%). The median duration of response (DOR) was not reached in patients with thymoma and 9.7 months in patients with TC [24]. Stable disease (SD) was observed in 5/7 patients with thymoma (71.4%) and in 14/26 patients with TC (53.8%). The median progression free survival (mPFS) was 6.1 months in both groups; median OS (mOS) was not reached in thymomas’ patients and 14.5 months in TC patients. Nevertheless, 5/7 patients with thymoma (71.4%) experienced at least one G3-4 irAE (two hepatitis, three myocarditis, one thyroiditis, one colitis, one conjunctivitis, and one nephritis), whereas severe irAEs were reported in only 4/26 patients with TC (15.3%) (two hepatitis, two MG, and one subacute myoclonus). Most patients experiencing a severe immune toxicity permanently discontinued immunotherapy and recovered with high-dose corticosteroids or other immunosuppressive agents, but one patient affected by thymoma died following an opportunistic infection developed during immunosuppressive treatment.

Pembrolizumab was tested in another trial enrolling 40 patients with TC. Tumor response (1 CR; 8 PR) and SD were observed in 9/40 (22.5%) and in 21/40 patients (52.5%), respectively. The median DOR was 36 months. The mPFS and mOS were 4.2 and 24.9 months, respectively [23,62]. The G3-4 AEs (transaminases increasing, dyspnea, myalgia or myositis, increased creatine phosphokinase) were detected after a median time of four cycles and occurred in 6/40 patients (15%). Also two cases of polymyositis and myocarditis requiring a pacemaker implantation were reported. Other severe irAEs included anemia, thrombocytopenia, arthralgia, hyperglycemia, blurred vision, and increasing lipase. High-dose steroids were delivered in 5/6 patients (83.3%), and two of them required additional immunosuppression, with resolution of the toxicities; one of the patients, developing type 1 diabetes mellitus, did not receive steroids, but needed a permanent insulin therapy.

A phase I trial is ongoing to assess the best dose and the toxicities of pembrolizumab as monotherapy in patients with unresectable thymomas and TCs who do not have pre-existing autoimmune diseases (NCT03295227) (Table 2).

Nivolumab was tested on 15 platinum-pretreated patients with TC in a single-arm phase II trial (PRIMER study) [63]. It was formally negative, because the primary endpoint (ORR) was not met in any of the patients at a preplanned futility interim analysis, and consequently, the enrollment was early interrupted. Nevertheless, a disease control rate (DCR) was observed in 11/15 cases (73.3%), with five patients maintaining SD for at least 24 weeks. Overall, mPFS and mOS were 3.8 and 14.1 months, respectively. Serious irAEs occurred in two patients (13%), including one G3 aspartate aminotransferase elevation and one G2 adrenal insufficiency, although they did not lead to a permanent treatment discontinuation.

Nivolumab, as monotherapy or combined with ipilimumab, a cytotoxic T lymphocyte antigen 4 (CTLA-4) antibody, was also evaluated in a single-arm, two-cohort, phase II trial (NIVOTHYM study) on previously treated patients with B3 thymoma or TC. The PFS rate at 6 months (PFSR-6) as determined by central review, according to RECIST 1.1 criteria, was the primary endpoint [64]. Among 49 patients eligible for the efficacy analysis in the cohort 1 (Nivolumab alone), the PFSR-6 was 35%, which was not sufficient to meet the primary endpoint. ORR and DCR were 14% and 67.3%, respectively. Median DOR was 162 days. mPFS and mOS were 6.2 and 21.3 months, respectively. Any-grade treatment-related AEs were registered in 44 patients (81.5%), with severe ones in 14 of them (25.9%). The most common G3-4 irAE was colitis, occurring in two patients (3.7%). The combination of nivolumab and ipilimumab is currently under evaluation in the second cohort of the study (Nivolumab + Ipilimumab).

Another phase II trial (NCT04925947), evaluating the dual blockage of PD1 and CTLA-4, is testing KN046, a bi-specific antibody against these two targets. In this study, KN046 efficacy and safety is assessed in patients with advanced TC progressed after at least one prior session of check-point inhibitor therapy.

Avelumab and Atezolizumab were tested in recurrent or refractory TETs. Avelumab was assessed in a phase I dose-escalation trial on eight patients (seven thymomas and one TC) [65]. SD was reported in patients with TC, while DC (four PR; two SD) was achieved in six cases with thymoma (85.7%). All responders experienced an irAE (three myositis and one enteritis), versus only one patient out of four non-responders. All of these patients were treated with high-dose steroids, also with intravenous immunoglobulin in one case, and permanently discontinued Avelumab. An ongoing phase II trial is evaluating Avelumab in patients with TETs after platinum-based chemotherapy (NCT03076554).

Atezolizumab was tested in a phase II multi-cohort study (also including a group of pretreated TETs), having the non-progression rate (NPR) at 18 weeks as the primary efficacy endpoint. The thymoma subgroup achieved the highest median NPR at 18 weeks (76.9%, n = 10/13 patients). PR was observed in 5/13 patients (38.5%), and the response duration was longer than 1.4 years for 4/5 patients. Nevertheless, this cohort had also the highest percentage of severe irAEs (35.7%), with one treatment-related death [66].

Several ongoing studies are evaluating different agents with ICIs, like tyrosine-kinase inhibitors (TKIs) or chemotherapy [69,70,71,72]. In a single-arm phase II study (CAVEATT trial), 32 pretreated patients (27 TC, 3 thymoma B3, and 2 mixed thymoma B3 and TC) were treated with Avelumab plus Axitinib. The ORR was the primary end point. PR was observed in 11/32 patients (34%), with a median DOR of 5.5 months. Overall, the DCR was 90%, with a median PFS and OS of 7.5 and 26.6 months, respectively. Hypertension was the most frequent severe AE, reported in 6/32 patients (19%) and related to Axitinib, while G3-4 irAEs occurred in four patients (12%), with three cases of G3-4 polymyositis, and one case of G3 pneumonitis [67].

Ongoing trials are testing the combinations of Sunitinib plus Pembrolizumab (NCT03463460), Lenvatinib plus Pembrolizumab (PECATI trial, NCT04710628), and Vorolanib plus Nivolumab (NCT03583086).

Immunotherapy plus chemotherapy is under evaluation. In a phase II trial, patients with treatment-naïve unresectable TETs (Masaoka stage III or IVA) are receiving 3 cycles of neoadjuvant Pembrolizumab combined with Docetaxel and Cisplatin, followed by surgery and 32 more cycles of adjuvant Pembrolizumab. In case of R1/R2 resection or in case of non-progressive but still unresectable disease after neoadjuvant treatment, radiotherapy will be associated with immunotherapy with Pembrolizumab for 32 cycles (NCT03858582).

Moreover, a phase II single-arm trial has been planned to evaluate the efficacy and safety of combining chemo-immunotherapy and TKI in untreated advanced or recurrent TCs. After a maximum of four courses with carboplatin, paclitaxel, pembrolizumab, and lenvatinib, a maintenance treatment with pembrolizumab and lenvatinib will be continued until progression or unacceptable toxicities (NCT05832827).

### 5.2. Cancer Vaccines

The identification of tumor-associated antigens (TAAs), whose epitopes are recognized by human leukocyte antigen (HLA) class I-restricted cytotoxic T lymphocytes (CTLs), has driven the development of cancer vaccines against several tumor types [73]. Nevertheless, because of a low average tumor mutational burden (TMB), only a small amount of neoantigens are expressed in TETs, making it difficult to identify potential targets [74]. A WT-1 peptide vaccine was evaluated in a phase II trial in patients with relapsed/refractory, HLA-A*24:02 positive thymoma, or TC expressing Wilms’ tumor-1 (WT-1) protein, which was found to be overexpressed in TETs (thymomas: 80%; TCs 84.6%) [68]. Among the 15 enrolled patients, 12 were evaluable for response (4 thymomas, 8 TCs): despite an ORR of 0%, a DCR of 75% was observed (6/8 TCs and 3/4 thymomas). WT-1 vaccination was well tolerated, without any immediate severe AEs, whereas 2 long-term treated patients affected by thymoma experienced an irAEs after more than 2 years of therapy (one G3 PRCA; one G2 MG).

### 5.3. Immunomodulatory Agents

Indoleamine 2,3-dioxygenase-1 (IDO1) is a catalytic enzyme responsible of tryptophan degradation into kynurenine. IDO1 plays an important role in immune regulation leading to the suppression of T cell response against antigens released by apoptotic cells. In cancer, this protein can be expressed by both malignant and myeloid-derived cells in the tumor microenvironment and favors cancer immune escape. Wei et al., using microarray analysis of tissue samples, observed a 13% and 14% IDO1 expression rate in thymomas and TCs, respectively, and a survival benefit in patients with low IDO1 and high FOXP3 Tregs expression levels [75]. Epacadostat, an IDO1 inhibitor associated with pembrolizumab, is under investigation in 45 TET patients progressed to at least one line of chemotherapy in an ongoing phase II clinical trial (NCT02364076). Preliminary results have shown an ORR of 22,5%, a DCR of 75%, and a median PFS and OS of 4.2 months and a median OS of 24.9 months. Serious AEs were registered in 30% of patients including myocarditis, hyperglycemia, hepatitis, bullous pemphigoid, and polymyositis [76].

Transforming growth factor-β (TGF-β) is another protein that plays an important role in cancer immune tolerance. This cytokine favors tumor growth and inhibits immune response against malignant cells, by reducing cytotoxic CD8 lymphocytes expansion and proliferation, suppressing cytotoxic protein transcription in T cells, and activating Tregs. In an immuno-histochemistry retrospective analysis of tissue biopsies from 33 patients with TETs (20 advanced TC and 13 thymomas), Duan et al. observed an OS advantage and a high CD8 expression in those with low TGF-β expression levels [77].

Bintrafusp alfa, a bifunctional fusion protein against TGF-β and PD-L1, is being tested in TET patients progressed after at least one platinum-containing chemotherapy in an ongoing, open label, phase II study (NCT04417660) [78].

In light of the early promising evidence of the activity of immunotherapy in TETs, current research is investigating novel combination strategies with the aim of maximizing therapeutic response. In particular, some early preclinical trials focused on the role of mitochondria in PD-L1 degradation via adenosine 5′ monophosphate-activated protein kinase (AMPK) activation. In these studies, nanoparticles were used to inhibit mitochondria-associated oxidative phosphorylation, thus favoring PD-L1 down-regulation by AMPK. In spite of such promising preclinical data, the validation of these new therapeutic strategies in clinical setting is needed to define their applicability in patients’ care [79].

## 6. Discussion

Considering the rarity of TETs, their management is still challenging. Unlike other solid tumors, the development of targeted biologic and/or immunologic therapies in TETs is still at the beginning. In this regard, the administration of anti-PD-1 inhibitors have been tested in a limited series of stage IV diseases. Despite promising clinical activity, the benefits of ICIs among patients with TETs should be carefully balanced with the increased risk of immune toxicity reported in clinical trials. Moreover, the association between TETs and ADs makes the administration of novel immunotherapies even more difficult, due to the higher risk of exacerbating autoimmunity in these patients.

The identification of predictive biomarkers is desirable in order to distinguish potential responders and non-responders to ICI-based therapies. High levels of PD-L1, defined as positive staining via immunohistochemistry (IHC) in ≥50% of tumor cells, seem to correlate with an improved response rate to Pembrolizumab and OS compared with low or absent PD-L1 expression [23,24]. However, these early data derive mainly from patients with TCs, and larger studies are needed to also confirm it in patients with thymomas. Furthermore, the PD-L1 expression via IHC was not predictive of response to Avelumab plus Axitinib [67]. In a phase I trial testing Avelumab, responders were found to have higher pre-treatment absolute lymphocyte count, higher baseline T cell receptor (TCR) diversity, and lower frequencies of B cells, Treg, natural killer (NK) cells, and conventional DCs [65]. A genomic profiling of patients with TC treated with Pembrolizumab revealed the presence of CYLD mutations in responders and BAP1 mutations in non-responders. Interestingly, these mutations seem to correlate with specific patterns of PD-L1 expression, although it remains to be investigated if the responsiveness of TC to Pembrolizumab is related to the genomic characteristics of the tumor or to the levels of PD-L1 [80]. Moreover, as no reliable biomarkers are available to date, the potential toxicity of ICIs in patients with TETs cannot be ignored. Patients with TETs receiving with PD-1/PD-L1 inhibitors are more likely develop musculoskeletal, neuromuscular, and myocardial immune-mediated toxicity in comparison with other types of cancer. In particular, patients with thymoma presented a high percentage of potentially severe irAEs compared to patients with TC (G ≥ 3 irAEs: 58.3% vs. 17.1%) [52,81,82,83,84]. In fact, MG has been reported as an irAE in 3–14% of patients with TET receiving Pembrolizumab, while myositis was observed in 8% of patients with TC treated with Pembrolizumab and in more than 50% of patients with thymoma receiving Avelumab. Myocarditis has been reported in 5% of patients with TC and in 43–57% patients affected by thymoma treated with ICIs [23,24,65,85]. Notably, as observed in other malignancies, and also in TETs, the development of irAEs has frequently been related with a disease response [23,65,86,87,88,89].

Which mechanisms underlie the increased susceptibility to immune-related toxicity in patients with TET is not clearly understood. It is probably that ICIs could interfere with the thymic epithelial cell death and consequently lead to the loss of immune tolerance and an increase in the rate of irAEs [55,90,91]. In order to identify biological features useful to predict and prevent autoimmune toxicity, an analysis of circulating mononuclear cells from patients with thymoma treated with Avelumab was performed. At baseline, patients experiencing irAEs had a higher absolute lymphocyte count, a higher degree of TCR diversity, and a lower level of B-cells, Treg cells, and DCs. Additionally, patients who developed treatment-related myositis had, before immunotherapy, detectable titles of acetylcholine receptor-binding antibodies [65,92].

## 7. Conclusions

Advancing our knowledge on the immune pathogenesis of TETs is crucial. In clinics, physician needs to know why some patients develop AD after thymectomy or vice versa or if there are patients at higher risk to develop ADs. Hence, a better understanding of the pathophysiological links between autoimmunity and TETs (especially thymomas) and the identification of potential predictive biomarkers of response to immunotherapy and the development of ADs could allow physicians to select patients at higher risk of developing autoimmune disorders, in order to prevent the onset of ADs and to avoid the immune-related side effects or activation/recrudescence of autoimmune disorders induced by immunotherapy, thus representing a promising therapeutic option for these rare tumors.

## Figures and Tables

**Figure 1 cancers-15-05574-f001:**
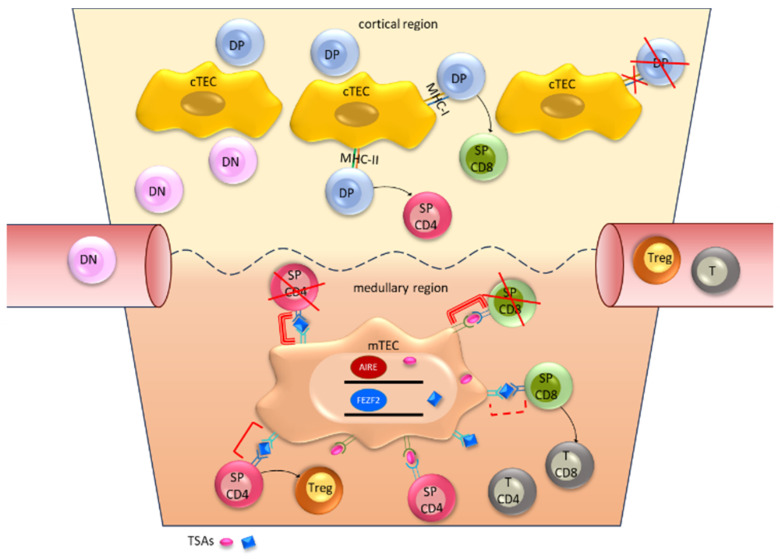
Circulating lymphoid progenitor cells, early T cells lacking both CD4 and CD8 expression (DN), migrate into thymic cortico-medullary junction. T cell precursors reach the cortical region via the expression of chemokine receptors following the gradients of their corresponding ligands produced by stromal cells. DN thymocytes initiate the rearrangement of the TCR locus of the β chain. Once selected, T cells simultaneously express CD4 and CD8 (DP thymocytes). Positive selection: functional TCR expressed on DP thymocytes interacts with MHC expressed by cTEC. Clones of thymocytes that recognize antigens on the MHC, receive a survival stimulus, while thymocytes that fail to recognize self-MHC undergo apoptosis (death by neglect). According to the class of MHC recognized, MHC-I or MHC-II, DP thymocytes commit to either CD8 or CD4 lineage, respectively. Completed positive selection, SP thymocytes migrate in the thymic medulla following a chemokine gradient secreted by mTECs. mTECs are characterized by the expression of AIRE. During negative selection, high-affinity interaction between TCR and self-peptide-MHC, in the presence of costimulatory signals, results in the activation of the apoptotic cascade. A fraction of CD4 thymocytes with strong MHC-II interactions transform into regulatory T cells (Treg).

**Table 1 cancers-15-05574-t001:** Immunotherapy and TETs.

Drug [Reference]	Design	N° Enrolled T/TC Patients	RR	mPFS (95% CI)	mOS (95% CI)	Rate of Severe irAEs
Pembrolizumab [24]	Phase II single arm	33 (7 T, 26 TC)	T: 28.6% TC: 19.2%	T: 6.1 m (4.3–7.9) TC: 6.1 m (5.1–7.1)	T: NR TC: 14.5 m	T: 71.4% TC: 15.4%
Pembrolizumab [62]	Phase II single arm	40 (TC)	22.5%	4.2 m (2.9–10.3)	24.9 m (15.5-NR)	15%
Nivolumab [63]	Phase II single arm	15 (TC)	0%	3.8 m (1.9–7.0)	14.1 m (11.1-NE)	20%
Nivolumab (cohort 1) [64]	Phase II single arm ^1^	55 (10 T, 43 TC)	12%	6.0 m (3.1–10.4)	21.3 m (11.6-NE)	57%
Avelumab [65]	Phase I dose escalation	8 (7 T, 1 TC)	T: 28.6% TC: 0%	NR	NR	83.3%
Atezolizumab [66]	Phase II single arm, multi-cohort, basket ^2^	13 (T)	38.5%	NR	NR	35.7%
Avelumab + Axitinib [67]	Phase II single arm	32 (3 B3-T, 2 B3-T/TC, 27 TC)	34% (B3-T and B3-T/TC: 40%; TC: 33%)	7.5 m (3.7–10.0)	26.6 m (17.0–30.0)	12%
WT-1 peptide vaccine [68]	Phase II single arm	12 (4 T, 8 TC)	T: 0% TC: 0%	NR	NR	T: 25% TC: 0%

^1^ Reported data refer to cohort 1 (nivolumab); cohort 2 (nivolumab and ipilimumab) is ongoing; ^2^ The reported data refer to the T cohort. Abbreviations: RR: response rate; DCR: disease control rate; mPFS: median progression free survival; mOS: median overall survival; T: thymoma; TC: thymic carcinoma; NR: not reported; B3-T: B3 thymoma; B3-T/TC: mixed type B3 thymoma and thymic carcinoma; pts: patients; m: months, irAEs: immune-related adverse events.

**Table 2 cancers-15-05574-t002:** Ongoing clinical trials evaluating immunotherapy in TETs.

Clinical Trial	Phase	Agent	Target	Intervention	Tumor Type	Primary Endpoint	Status
NCT04667793	II	Toripalimab	PD-1	Neoadjuvant toripalimab + chemotherapy in locally advanced TETs.	T/TC	AEs, MPR	Recruiting
NCT06019468	II	Envolizumab	PD-L1	Neoadjuvant envolizumab plus RT in TC	TC	ORR	Recruiting
NCT03134118 NIVOTHYM	II	Nivolumab	PD-1	Nivolumab in platinum-progressive T/TC	B3-T/TC	PFS	Active, not recruiting
NCT04417660	II	Bintrafusp alpha (M7824)	PD-1/TGF-β	M7824 in platinum-progressive T/TC	T/TC	ORR	Recruiting
NCT05104736	II	PT-112	Epithelial cells	PT-112 in advanced T/TC	T/TC	ORR	Recruiting
NCT04710628	II	Pembrolizumab	PD-1	Pembrolizumab + lenvatinib in pretreated T/TC	B3-T/TC	PFS	Recruiting
		Lenvatinib	VEGFR/PDGFR				
NCT03583086	I/II	Nivolumab	PD-1	Vorolanib + nivolumab in TC	Thoracic tumors (incl. TC)	ORR, DOR, PFS; DCR, 1-year OS	Active, not recruiting
		Vorolanib	VEGFR/PDGFR				
NCT03463460	II	Pembrolizumab	PD-1	Pembrolizumab + sunitinib in platinum progressive TC	TC	ORR	Recruiting
		Sunitinib	VEGFR/PDGFR				
NCT04234113	I/Ib	Pembrolizumab	PD-1	SO-C101 ± pembrolizumab in advanced solid tumors	Advanced solid tumor (incl. T/TC)	AEs	Active, not recruiting
		SO-C101	IL-15				
NCT05544929	I	Tislelizumab	PD-1	KFA115 ± tisletizumab in selected advanced cancers	Select advanced cancers (incl TC)	AEs	Recruiting
		KFA115	Helios/IKZF2				
NCT03295227	I	Pembrolizumab	PD-1	Pembrolizumab in advanced TET	T/TC	Safety/tolerability	Recruiting
NCT03556228	I	VMD-928	TrkA (NTRK1)	VMD-928 in advanced solid tumors	Advanced solid tumor (incl. T/TC)	AEs	Recruiting

Abbreviations: AEs: adverse events; B3-T: B3 thymoma; B3-T: type B3 thymoma; DCR: disease control rate; MPR: major pathologic response; NTRK1: neurotrophic tyrosine receptor kinase; ORR: objective response rate; OS: overall survival; PFS: progression free survival; PD-1: Programmed cell death protein-1; PDGFR: platelet-derived growth factor receptor; T: thymoma; TC: thymic carcinoma; TET: thymic epithelial tumors; TGF-β: transforming growth factor-β; VEGFR: vascular endothelial growth factor receptor.

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
