# Peer review of "Thymic Epithelial Tumor and Immune System: The Role of Immunotherapy"

_cancers, 2023, doi:10.3390/cancers15235574_

Round 1

Reviewer 1 Report

Comments and Suggestions for Authors

This manuscript presents a clear, engaging, and critical overview of the challenges and opportunities for immunotherapeutic strategies for thymic epithelial tumors. Addressing potential future directions would further enhance the overall impact and completeness.

Author Response

Thank you for your comments.

Reviewer 2 Report

Comments and Suggestions for Authors

There are markers of this tumor spread?  Has radiotherapy a role?

Author Response

Thank you for the comments. The answers to your criticisms are reported in the attached file.

Reviewer 3 Report

Comments and Suggestions for Authors

In this research, the authors reviewed the thymic epithelial tumor and immune system: the role of immunotherapy. In my opinion, the current version of this manuscript fits the scope of Cancers and could be accepted after minor revision.

My specific comments are in detail listed below:

1.     Some references are out of date (published before 2010). The authors should revise it. Some new references could be accepted.

2.     In this part ‘Role of immunotherapy’ (Line 267-367), apart from the commonly used PD-L1/PD-1 inhibitors. The recent development of small molecular inhibitor or regulators in thymic epithelial tumor treatment and some other related tumors. Some references should be added to this part including 10.1016/j.jconrel.2022.11.004.

3.     Did some other immune checkpoints were affected after thymic epithelial tumor therapy, like Cox-2, TGF-b, CD24, CD47?

4.     In the discussion part, the authors could predict how mitochondria affect the therapy efficacy of thymic epithelial tumor and some other related tumors. Some references should be added to this part including 10.1002/adma.202206121.

5.     In conclusion, the clinical transformation barriers of immunotherapy in treating thymic epithelial tumor should be better out-looked.

Author Response

Thank you for your comments. The answers to your criticisms are reported in the attached file.

Reviewer 4 Report

Comments and Suggestions for Authors

Matteo et al. summarized lots of literature about the thymus’s immune biology and its association with autoimmune paraneoplastic diseases, as well as the results of the available studies with immune checkpoint inhibitors and cancer vaccines. They spent a lot of effort reading, sorting, and summarizing the literature. I think it will make a considerable contribution to the development of the field. From my point of view, the language should be more concise. If there are more figures and tables, it will be easier for readers to understand.

Author Response

Thank you or your comments. The answer to your criticisms are reported in the attached file.
